# "Dispense antiretrovirals daily!" restructuring the delivery of HIV services to optimize antiretroviral initiation among men in Malawi

**Alinane Linda Nyondo-Mipando**[1]*, **Leticia Suwedi Kapesa**[1], **Sangwani Salimu**[1], **Thokozani Kazuma**[1], **Victor Mwapasa**[2]

**1** Department of Health Systems and Policy, School of Public Health and Family Medicine, College of Medicine, Blantyre, Malawi, **2** Department of Public Health, School of Public Health and Family Medicine, College of Medicine, Blantyre, Malawi

* lmipando@medcol.mw

**Data Availability Statement:** Relevant data are within the paper. However full access to the data is restricted to available upon request per ethical

## Abstract

### Background

Gender disparities exist in the scale-up and uptake of HIV services with men being disproportionately under-represented in the services. In Eastern and Southern Africa, of the people living with HIV infection, more adult women than men were on treatment highlighting the disparities in HIV services. Delayed initiation of antiretroviral treatment creates a missed opportunity to prevent transmission of HIV while increasing HIV and AIDS-associated morbidity and mortality. The main objective of this study was to assess the strategies that men prefer for Antiretroviral Therapy (ART) initiation in Blantyre, Malawi.

### Methods

This was a qualitative study conducted in 7 Health facilities in Blantyre from January to July 2017. We selected participants following purposive sampling. We conducted 20 in-depth interviews (IDIs) with men of different HIV statuses, 17 interviews with health care workers (HCWs), and 14 focus group discussions (FGDs) among men of varying HIV statuses. We digitally recorded all the data, transcribed verbatim, managed using NVivo, and analysed it thematically.

### Results

Restructuring the delivery of antiretroviral (ARVs) treatment and conduct of ART clinics is key to optimizing early initiation of treatment among heterosexual men in Blantyre. The areas requiring restructuring included: Clinic days by offering ARVs daily; Clinic hours to accommodate schedules of men; Clinic layout and flow that preserves privacy and establishment of male-specific clinics; ARV dispensing procedures where clients receive more pills to last them longer than 3 months. Additionally there is need to improve the packaging of ARVs, invent ARVs with less dosing frequency, and dispense ARVs from the main pharmacy. It was further suggested that the test-and-treat strategy be implemented with fidelity and revising the content in counseling sessions with an emphasis on the benefits of ARVs.

guidelines imposed by Malawi's College of Medicine Research and Ethics Committee. Datasets can be made available at reasonable request from the corresponding author in compliance with College of Medicine's requirements. Interested researchers may also contact The Postgraduate Dean is the Director of Research as well and custodian of College of Medicine's generated data at postgraduate@medcol.mw.

**Funding:** This study was funded by the Malawi HIV Implementation Research Scientist Training program (Fogarty: D43 TW010060).

**Competing interests:** The authors have declared that no competing interests exist.

## Conclusion

The success in ART initiation among men will require a restructuring of the current ART services to make them accessible and available for men to initiate treatment. The inclusion of people-centered approaches will ensure that individual preferences are incorporated into the initiation of ARVs. The type, frequency, distribution, and packaging of ARVs should be aligned with other medicines readily available within a health facility to minimize stigma.

## Introduction

Gender disparities continue to exist in the scale-up and uptake of Human Immunodeficiency Virus (HIV) services with men disproportionately represented in accessing the services [1]. Although heterosexual men are active transmitters of HIV infection, they are regarded as less vulnerable to HIV with limited attention in preventive efforts and policies [2, 3] and at best have been stereotyped as the ones that spread the virus [4]. Despite the limited attention that heterosexual men receive concerning HIV services, heterosexual transmission remains the primary mode of contracting HIV in Sub-Saharan Africa (SSA) [5]. Although several studies have documented the benefits of starting ARVs early regardless of Cluster of Differentiation 4 (CD4) count [6–8], men have lagged in the uptake of Antiretrovirals in SSA [9, 10]. The recent trends in Malawi also show a similar pattern where 92% of HIV-infected women were on treatment while men lagged at 76% [10]. As illustrated in several studies, men tend to access HIV services at a late stage [11–13] and consequently delay initiating ARVs [14]. A delay in the initiation of ARVs creates a missed opportunity to prevent transmission of HIV [12] and increases the mortality rate in men consequent to initiating ART in an advanced AIDS state [11, 15].

Among the factors contributing to low uptake of ART services by men is poor knowledge of the availability of services, perception of having a low risk [16] limited avenues for HIV services for men [17], fear of discrimination, stigma, and rejection, and [18] notions of masculinity that present men as resilient to illness [19]. Furthermore, the health system has been criticized for being uninviting to people living with HIV which is compounded by its structures that are authoritarian in the delivery of services [20]. Additionally, it has been argued that the services are biased towards females [21] which has resulted in limited services for men. The services that are available for men have primarily focused on men having sex with men leaving out the general population of men [22, 23].

Several strategies have been proposed to address inequalities among men. Shand et al. [24] advocate for challenging the patriarchal structures that contribute to the notion of resilience among men which makes them seek medical services late. New strategies within the health system that aim at optimizing service delivery to various clients are called Differentiated Service Delivery (DSD) models. Differentiated Service Delivery models are advocated as a framework for health systems to deliver services that consider the user's preference [25, 26]. Key to the DSD models are the four tenets that can be applied and these include a variation in the types of services provided, varied locations where the service will be provided, a range of health care workers providing the service, and the frequency of services provided [25]. Each tenet addresses a specific need that a client may prefer for accessing ARVs [25]. The main advantage of DSD models is that they optimize usage of the limited number of staff available in fragile health systems [27]. Differentiated Service Delivery models are cost-efficient [28, 29] and also reduce the need for more personnel [29].

Malawi adopted the DSD models in the delivery of HIV services in 2008 to optimize the delivery of HIV-related services [28]. HIV and ART services in Malawi are provided for free in all Government-owned health facilities and at a cost in private and faith-based organisations. The services include HIV testing, ART initiation, and follow-up in care. Health facilities in the urban areas offer HIV services daily while those in semi-urban and rural areas have a designated day within the week when ART clinics are run while HIV testing is offered daily. Malawi has integrated the delivery of HIV services with Tuberculosis, Family planning, and Sexual and reproductive health programmes [30]. All services are standardized and follow the same monitoring plan regardless of their location and rarely do facilities experience stock-outs of ARVs [31]. At the point of ART initiation, a client is supposed to nominate a person who is commonly referred to as a "guardian" who is usually a layperson and could be a friend or family member [32]. A guardian is counseled along with a client on the HIV status of the client and information on ARVs to promote adherence and at times to collect the ARVs on behalf of the client [32]. Despite the lag in men accessing HIV services, there have been no specific HIV service programmes rolled out for men in Malawi. However, anecdotal reports show that some health implementing partners piloted male-friendly clinic services in 12 primary health facilities in Blantyre, Malawi, in 2018, a year after we had finalized data collection [33]. The male-friendly clinics would open once a week on Saturday's targeting newly diagnosed men and those that had problems collecting ARVs during the week [33]. Male- friendly clinics were run by both female and male health care workers providing non-communicable health services, sexually transmitted infection services, HIV testing services, and ARVs dispensation [33]. Seeing that men have lagged in ART initiation, this study assessed strategies that men prefer for ART initiation in Blantyre, Malawi.

## Methods

### Study design

We conducted a descriptive qualitative study to assess strategies that men prefer for ART initiation in Blantyre, Malawi. We used In-Depth Interviews (IDIs), Key Informant Interviews (KIIs) [34, 35], and Focus Group Discussions (FGDs) [36]. We opted for IDIs because of the depth they guarantee in understanding a social phenomenon [34]. We used FGDs to stimulate varying responses from men secondary to the interactions within a group [37] and we employed aspects of a phenomenological approach by including men who have interacted with HIV services.

### Study setting

The study took place in Government-owned facilities and included six health centres and a tertiary hospital in Blantyre District in the southern part of Malawi. Of the six health centres; two are located in a rural area, two are in an urban area and the other two are in the semi-urban area. The two urban health centres are located in high densely populated areas in Blantyre city [40]. We also purposively selected the tertiary hospital because it is the main referral centre for Blantyre district and Southern Malawi. Blantyre district has the highest prevalence rate of HIV among men rate at 14 as of 2016 [38] which translates into more men needing ARVs.

### Sample size

We purposively [39] sampled our participants and included 20 men in IDIs and these were of the unknown, infected, and uninfected HIV statuses. We conducted 17 KIIs with Health Care Workers (HCWs) involved in the provision of HIV services at various intervals of HIV care

and these included HIV Diagnostic Assistants (HDAs), HIV Testing Services Providers, Medical Assistants, Clinical Officers, and Nurses. We also conducted 14 FGDs with men and we had 2 FGDs at each site. Our sample size was adequately guided by Guest who contends that saturation is mostly reached by the 12[th] interview [40]. The FGDs were divided according to age and translated into one FGD with younger men with an age bracket of 18–24 years and the other with older men with an age bracket of 25 years and above at each site. The FGDs were segregated according to age to promote participation which could have been limited if we had combined younger and older men in one group [37]. In total, we had 113 men in the FGDs. We included men of various demographics such as age, residency, education level, marital status, religion, occupation, and HIV status in our sampled participants, to achieve maximum variation so as to broaden the scope of the sources of information in our study [42].

## Identification, recruitment, and selection of study participants

**Identification of health care workers.** After explaining the study and the eligibility criteria, the Officer In-charge assisted with the identification of health care workers and the selection was based on their role in the provision of HIV services to men. For the HCWs, we included only those that had provided ART services for more than 6 months at the selected site and were available and willing to participate in the study. The Blantyre District Health Office (DHO) referred us to the Blantyre District's HIV and ART Coordinators for inclusion in the study. None of the health care workers approached refused participation in the study.

**Identification of participants for in-depth interviews.** We purposively identified men for in-depth interviews following the study's eligibility criteria. We included heterosexual men (as ascertained by asking them their sexual orientation) who were 18 years of age and above, able and willing to provide consent, and of a range of HIV statuses: uninfected (confirmed through checking the health passport books), unknown (ascertained by asking if they have ever tested for HIV or not), and HIV infected on ART and not on ART (confirmed through checking the health passport book and/or clinic records). If a man was interested in the study we either booked an appointment with him on a later date or if flexible study procedures were conducted after completion of the primary aim for visiting the health facility. Participants who were not on ARVs were identified with assistance from health care workers who introduced them to research staff. Five participants that were approached refused participation and cited time constraints and lack of interest in the study as the reasons for non-participation.

**Identification of participants for focus group discussions.** We identified participants for FGDs with assistance from HCWs following the eligibility criteria. We included men that were 18 years and above, varying HIV status, on ARVs and not on ARVs, able to provide consent, and willing to participate in the study. Men were scheduled for a discussion at a time and place convenient for them. Seven participants that were approached refused participation and cited time constraints as the reason for non-participation.

## Data collection

We collected data from January to July 2017 using pretested interview and discussion guides that were developed based on the study objectives and literature. All interviews and discussions were conducted once-off per participant or discussants respectively. With the exception of VM, all authors conducted interviews and all data collectors were female. Before data collection, ALNM, the Principal Investigator (PI), trained the team on the study protocol, procedures, research ethics, and logistics of the study. LSK and SS are public health specialists with Masters in Public health. TK has a Bachelor of Nursing and Midwifery and ALNM has a PH. D. in Health Systems and Policy. LSK was working as a Nursing Officer at the time of data

collection, SS and TK were Research Assistants in the project and ALNM worked with the College of Medicine. There was no prior relationship with the participants and the data collectors introduced themselves as research assistants on the project while ALNM introduced herself as the Principal Investigator for the study. The study participants were informed that the rationale for conducting the study was to generate strategies for improving the initiation of ART among men in Malawi. All interviews and discussions were face-to-face and were conducted at a place and time convenient to the participants. All FGDs, KIIs, and IDIs were conducted at the respective health facilities except those of the ART and HIV Testing Coordinator which were conducted in their offices. The broad questions that guided the interviews and discussions for both men and health care workers were as follows:

1. Explain to me in detail the avenues or methods that can be used to increase early antiretroviral initiation among men at this facility and other areas?

2. Amongst the suggested strategies or avenues you have mentioned, which one would be most preferred by men [you] for early initiation of ARVs and why?

After a response, we probed further to capture more depth on the subject. Interviews and discussions with men were conducted in Chichewa while key informant interviews were deliberated in both English and Chichewa as per participant's preference. We captured all deliberations using a digital audio recorder and each recorded interview was assigned a unique identification number. During the interviews and discussions, we captured field notes that supplemented the audio recordings [41]. All audio records and completed transcripts were stored on a password-protected computer. We probed deeply on each response to ensure we covered adequate depth and scope on an issue. We conducted the interviews and discussion in the language a participant preferred in order to eliminate the language barrier which would compromise our findings. Each interview lasted for 30–45 minutes while FGDs lasted for about 60–90 minutes. Our decision to stop data collection was guided by the concept of saturation of ideas which was reached when we noted the absence of new ideas and responses from participants. We conducted and reported our study following the consolidated criteria for reporting qualitative research (COREQ) guidelines [42].

## Quality of data

We applied several measures to maximize the quality of our findings. We summarized the key findings after each interview or discussion to ensure that our findings are credible [43]. We described the methods used in conducting the study to ensure that our findings are dependable [43]. Furthermore we have provided a rich description of the setting and context where we conducted the study to make our results transferable to other areas [44].

## Data analysis

We used NVivo to manage our data and employed a thematic analysis approach as outlined by Braun and Clarke [45]. Data analysis commenced during the data collection period. All audio recordings from interviews and discussions were transcribed verbatim and simultaneously translated if the interview was not done in English. Codes were generated, deductively, from the objectives of the study and also, inductively, from the data [46]. First, the Principal Investigator listened to the audios and read the transcripts multiple times to gain a deeper understanding of the issues raised. Secondly, a codebook was generated and discussed among the research team for consensus. ALMN and SS independently coded separate transcripts to assess the applicability of codes. Any areas of disagreement were resolved through iterative discussions between ALNM and SS [45]. Thirdly, SS coded all the transcripts, and as she coded she

constantly discussed with ALNM the new codes to be added and any proposed changes to the codebook. Our three main themes were realized through the collation of similar, related, and recurrent codes, specifically, we grouped all codes on drugs such as dispensation, composition, packaging, and innovations under one overarching theme of differentiating drug dispensing modalities; all codes related to clinic flow, layout and hours and days of operation of a facility were aggregated under Differentiated clinic operations and all codes that referred to the implementation of test-and-treat and counseling were grouped under the strengthening implementation of available policies theme. Fourthly, the research team reviewed the themes and refined them as appropriate to achieve a correct representation of our findings [47]. Lastly, we reviewed the themes to ensure that they correctly represented the data under them without losing the meaning of the data.

## Ethical considerations

Our study obtained ethical approval from Malawi's College of Medicine Research and Ethics Committee (COMREC- Number P.11/16/2064) before study implementation. The District of Health and Social Services (DHSS) for Blantyre and the Director of (Queen Elizabeth Central Hospital (QECH) provided Institutional support for the study to take place in the respective sites. Each study participant provided written informed consent before any study procedures. Participants that could not read nor write thumb-printed on the consent form after it was read to them in the presence of an impartial witness. All audios and transcripts were saved in a password protected computer with limited access to the researchers.

## Results

### Characteristics of health care workers

Fifteen of the 17 health care workers were male with a median age of 38 years old and an interquartile range (IQR) of 32–48 years old. Fourteen were married and 9 were HDAs, 4 were Nurses, 1 was a Medical Assistant, 2 were ARV Coordinators and 1 was an HIV Testing Services Coordinator.

### Characteristics of men in the study

The median age of the men was 27 [IQR 21–35]. Most men were married and had attained a secondary school education. More men were self-employed and of the 40 that were HIV infected, 32 were on ARVs (Table 1)

### Restructuring the delivery of ART services

Participants suggested restructuring the delivery of services in the following areas: Clinic Operations, drug dispensation, and implementation of HIV and AIDS policies. Table 2 below summarizes the key findings. These were proposed to ensure that privacy is maintained to avert undue disclosure of an HIV infected status which may result in being stigmatized.

  **1. Differentiate clinic operations.**   *1.1 Clinic days*: *Offer ARVs daily*. Participants stated that ARVs should be dispensed daily to avoid unintended disclosure of an HIV positive status when they are dispensed on designated days. This was commonly stated among all men and health care workers from rural and semi-urban health centres where ARVs are not dispensed daily. It was stated that on the day when ARVs are dispensed in the rural and semi-urban health centres, all other activities like clinics cease from functioning because the focus for the day is usually on the ART clinic such that community members can easily conclude that all those available at the clinic on such a day are likely to be HIV infected. Participants stated that

**Table 1. Characteristics of men that were involved in the study.**

| Variable | Number | Percentage (%) (N = 133) |
|---|---|---|
| Age | 27 (IQR 21–35) | |
| Marital Status | | |
| • Married | 65 | 48.87 |
| Literacy | | |
| • Able to read | 120 | 90.22 |
| Education Level | | |
| • No education | 7 | 5.26 |
| • Primary | 41 | 30.83 |
| • Secondary | 69 | 51.88 |
| • Tertiary | 16 | 12.03 |
| Employment | | |
| • Not Employed | 52 | 39.10 |
| HIV Testing^ | | |
| • Had an HIV Test | 101 | 77.10 |
| • HIV Infected | 40 | 39.60 |
| • HIV Uninfected | 61 | 60.40 |
| Uptake of ARVS^ | | |
| • On ARVs* | 32 | 80.00 |

*- The denominator is those that are HIV infected

^- These sections will not add up to N because they are a sub-set from N

this attracts undue disclosure of an HIV status which most men feared may result in stigma and discrimination.

> *"Dispense ARVs daily. This can help people to access ARVs daily because it happens that most of the time people cannot manage to come on Wednesday, just for ARVs . . ..but the clinic should be open on all days."* Younger men FGD Participant at Centre 2

Stigma was a salient factor behind the suggestion to dispense ARVs daily. Men of unknown and known HIV status believed that an important factor for men to initiate on ARVs is having services that are not stigmatizing. They further stated that the current services are prone to stigmatization because of the designation of ART clinic days. Health care workers had similar sentiments of averting stigma and corroborated the men's suggestion of providing ARVs daily.

**Table 2. Summary of key findings.**

| Strategy | Areas to restructure | Categories |
|---|---|---|
| Restructuring of ART service delivery | Differentiate clinic operations | Clinic days: Dispense ARVs daily |
| | | Clinic hours: extension of operational hours |
| | | Clinic layout and flow (Male-specific clinics) |
| | Differentiate drug dispensing modalities | Offer more drugs |
| | | The invention of ARVs with less dosing frequency |
| | | Dispense ARVs in the absence of a guardian |
| | | Dispense ARVs from the main Pharmacy |
| | | Packaging of ARVS |
| | Strengthen implementation of policies | Strengthen implementation of test-and-treat Strategy |
| | | Reinforce counseling that emphasizes benefits of ARVs |

"The other thing that increases discrimination is the practice of giving out ARTs on specific days only. So on that particular day, people know that those who are here (health facility) today are here to get ARVs." HIV Uninfected man, IDI at Centre 3

*1.2 Clinic hours*: *Extension of office hours*. Participants suggested that ART Clinics open for longer by opening on the weekends and closing late in the evening during the weekdays. This change if implemented will accommodate the demanding work schedules of men which is a deterrent to ART initiation. ART services are usually offered from morning till 4 pm on weekdays only.

*"At least we should have a health worker who will be able to work even at lunch hour so that other people should come at that time to get the drugs . . . it would have been better for people to be able to get the drugs even during the night may be up to 9 pm."* KII, HDA at Centre 6

Upon reflection on their operations and in light of the proposed hours of operation, healthcare workers recommended offering a client the various options of ART initiation available at a facility so that a man makes an informed decision on how and when to access ARVs. Consequently, a man will have a choice to select whether to attend the normal working hours, evening, or weekend services which will promote the initiation of ARVs because a man would select the option most convenient for him.

*1.3. Clinic layout and flow*. Men of unknown and known HIV statuses complained of the current layout and flow of the clinics where HIV testing and ART occurs. They stated that the rooms for HIV testing in all centres are located in separate locations from where ARVs are dispensed and this compromises their privacy. In a quest to maintain privacy and avoid indirect disclosure of an infected status, men suggested creating a flow that is easy to follow by having HIV testing rooms in the same location as the ARVs initiation section.

*"They (health care workers) have to change their system and the layout of this hospital in the sections where they conduct HIV testing and provide ART, these sections should be close to each other and the places should also be secure so that these men should be comfortable to come here and access these services".* Younger Men FGD Participant at Centre 5.

Furthermore, men complained of waiting for a long time at the facility before getting all the assistance they need. They recommended a reorganization of the services to hasten a quick exit from the clinic and they believed that this will encourage more men to initiate ARVs.

*"They must not take long to get the drugs at the hospital because if they take long to get the drugs then one can decide to just go and leave those drugs without accessing them."* Man not on ART at Centre 5

As a measure of managing waiting times and safeguarding the privacy of men, both health care workers and men suggested the creation of male-specific clinics that have a designated day and time when services are provided. They asserted that male-specific clinics will promote attendance by men because they will not be mixed with women as it is currently done which is a deterrent for timid men.

*"Maybe we should arrange a date or a day for males as we do with women on family planning days or under-five clinics, so that they should come and receive different services such as male circumcision, HIV testing, and AIDS counseling."* KII, HDA at Centre 2

**2. Differentiated methods of dispensing drugs.** *2.1 Offer drugs to last for a longer period.* Participants suggested that health facilities should dispense drugs that may last for a longer period than the current 1 to 3 months' supply so that they reduce the number of visits they make to the clinics for refills. The dispensation of drugs that last for a longer period would avert defaulting from treatment that is influenced by the frequency of visits and distance that one has to cover to access health services. Again, men further stated that their decision to initiate on ARVs is at times made after contemplating on the number of visits they have to make cognizant of the appointment schedules that are defined by a health facility.

*"Let us say the hospital is far from where you stay and when you go to the hospital they give you medication only for one month, and you should be going there every month, then you get discouraged. You think that all my life like 30 years, so how many trips will I make? They give a few medications but if they can give medication for maybe 5–6 months so that you can rest and go back after sometime."*–Younger Men FGD Participant at Centre 7

*2.2 Invention of ARVs with less dosing frequency.* Participants suggested the invention of ARVs with less dose frequency because some men delay initiation of ARVs due to the burden of taking them daily. To achieve a reduction in the frequency of taking drugs, health care workers suggested the invention of injectable ARVs that can be administered at spaced-out intervals or tablets that are not taken daily.

*"Had it been we can use the injection method it means that the person cannot take much of our time unlike where we are supposed to write everything because the client may take a lot of time. I have to write the next day of the appointment then take the drugs and give it to him unlike when we are supposed to use the injection because we will just give the client the injection and then later just use the registration, then off he goes and it will be a great and simple way".* KII, HCW at Centre 3

*"If there was one pill that should take a long time, three months, four months. Not just today, tomorrow, daily. But just to make one pill, when you take it, you should stay for some months."* Older men FGD Participant at Centre 4

*2.3 Offer drugs in the absence of a guardian.* Some participants argued that health facilities should initiate ARVs among men who have shown up without a guardian.

*"For the part of taking medicine early, I have seen other people sent back to get a guardian. And if the health workers can stop that and give medication to everyone at the time they come [without a guardian]."* Older Men FGD Participant at Centre 7

*2.4 Dispense ARVs from the main pharmacy.* Participants recommended that ARVs should be dispensed from the main pharmacy of a health facility to minimize stigma. They reiterated that the current practice of dispensing ARVs from a designated room heightens stigma because it indirectly discloses the status of a man who is seen around that room.

*"If we can start by giving these drugs through the same way at the pharmacy like what we do with everyone else who has any type of disease. . . if we keep on isolating them by telling them to come on their special day to get the ARVS, it will make a lot of people talk about them . . ."* KII, HDA at Centre 3

*2.5 Packaging of ARVS.* Participants suggested a revision in the packaging of ARVs to avoid unintended disclosure secondary to the conspicuousness of the drug bottles. Unlike other drugs that are dispensed in pill bags, ARVs are packaged in bottles that are difficult to conceal from others thereby disclosing one's status. Although health workers usually advise clients to purchase a carrier bag after exiting the clinic, men were still worried that unintended exposure would have already occurred at that point. Further to that, men stated that even if the bottles are in a bag, the pills in the bottle rattle, and others can easily notice that one is carrying ARVs and conclude that one is likely HIV infected.

*"Even if they say that I should buy the carrier bag or the plastic bag outside, people will be able to recognize them so they have to change and make something special so that people should not be able to recognize that you have carried bottles of ARVs in your hands."* Younger Men FGD Participant at Centre 5

**3. Strengthen implementation of policies.** *3.1 Strengthen implementation of test and treat strategy.* Health care workers and men reiterated the strengthening of the test-and-treat strategy because it has the potential of ensuring that men initiate ARVs instantly. In other cases, following and HIV infected test result, men are allowed to take time to discuss with their significant others before they initiate ARVs which leads to non-linkage to ART. One health care worker reported as follows:

*"During the test-and-treat strategy, we should not wait in initiating a man on ARVS, once the client is found to be positive, immediately he should start the treatment and then being counseled properly right there and not waiting for the guardian to be there".* KII, ART Coordinator

Health care workers noted that when facilities conduct outreach clinics in difficult-to-reach areas, they only conduct HIV tests without initiation of ARVs which is a missed opportunity for men to start on ARVs. As such, they recommended that ART initiation should be part of the services offered during outreach clinics. This will result in more men initiating on ARVs without having to travel a long distance to access ART at a facility.

*"It is good to work as a team by coming up with outreach clinics whereby the main focus will be the men but then we will go there with the HDA, the ART provider, and the doctor and then anyone who will be willing to access the HIV testing will be able to access ARVs on the same spot"* KII, HDA at Centre 3

*3.2 Reinforce counseling that emphasizes the benefits of ARVs.* With the advancement in the ART regimes, men and health care workers stated that the benefits of ARVs should be emphasized during counseling sessions to dispel the old belief that equated HIV infection to death. In the past, Malawi experienced many AIDS-related deaths which created fear around HIV and AIDS however, that trend has improved with the invention of better ARVS. Despite that positive trend, participants noted that the messages in the community have not trickled down to all for them to appreciate the relevance of initiating ARVs immediately

*"I think we need to be telling them properly about what happens when HIV is in the body, they can understand. If they understand then, they cannot wait to get sick before they start taking medication daily. . .. they don't know how the ARVs work in the body."* KII, HCW at Centre 4

Men emphasized that counseling and education should be specific to the individual and tailored to address individual issues rather than sharing a general health talk on HIV and AIDS.

*"Health workers should explain to every man clearly on his problem; because when a person is dying, he becomes very stubborn and adamant that he does not listen so tell everyone that if you have been found in this status, you are supposed to do ABC."* HIV infected man at Centre 4

## Discussion

Restructuring of the delivery and conduct of ART clinics is key to optimizing the early initiation of ARVs among heterosexual men in Blantyre. Participants in our study recommended restructuring in the following areas: clinic operations, drug dispensing procedures, and implementation of HIV and AIDS policies. All the concepts on restructuring services centered on reducing stigma and avoiding unintended disclosure that the health system imposes on the men. The maintenance of privacy and non-disclosure of an HIV status is the platform for the suggested ways of restructuring HIV services to promote early initiation of ARVs [47, 48]. Thus, the reduction of stigma is key to engagement with HIV services [49] particularly among men who are the most affected by the denial of an infected HIV status [50]. As a result of stigma, men refrain from engaging with HIV testing, initiating ART and retention in care [51, 52].

The dispensation of ARVs daily without designating a specific day as reported in our study is a measure to avert undue disclosure of an HIV status and courtesy stigma [53]. Stigma has been well documented as the main deterrent to engagement with HIV services at the different cascades [50, 54] and it's worsened by the location of the HIV/ART clinic [55] and the structural aspects of a vertical clinic [52]. Another disadvantage of offering ARVs on designated days is the promotion of congestion in the clinics that results in an unintended disclosure of one's HIV status [27]. To mitigate stigma, other countries advocate for pharmacy-only refills which entail dispensing of ARVs to stable clients and it allows for flexibility in accessing drugs [27]. Malawi could model pharmacy-only-refills for stable clients.

Having flexible operating rules of ART clinics is a step towards diffusing the prescriptive nature of clinics which disregards the preferences of the clients thus inhibiting the initiation of ARVs [20]. As alluded to earlier, integration of services would be a step towards the implementation of flexible operating rules. Services could be integrated through the adaptation of differentiated service delivery models [27] which have since been rolled out and are effective in Malawi [28, 56]. Health systems strengthening such as the definition of a stable patient is a prerequisite if a facility has to benefit from implementing the models [28, 56]. As HIV care moves into long term services, integrating them in the health system with other services will promote sustainability and utilization [57, 58]. Successful integration of HIV services in the routine provision of other clinic services will require an assessment of staffing levels [59] cognizant that resource-constrained countries may not be ready for effective integrated services [52].

Furthermore, our findings support service integration by suggesting revisions in the clinic flow which resonates with what was earlier reported in Zambia and South Africa [60]. The study reported that health care workers bemoaned distinct and demarcated areas for HIV services because they promote unintended disclosure and stigma hence affecting engagement with HIV services [60]. Our findings reiterate the results from a review that recommended alteration in the operations of a clinic like improving clinic management and integration of services as a measure to optimize initiation [61]. We argue that One-stop centres for all

services are an example of a service integration model that could be rolled out to promote engagement with HIV services among men. One-stop centres curb fragmented services that are conducive for unintended HIV status disclosure [60].

Our study also recommends that health facilities should extend the hours of operation of a clinic to accommodate men that may not be able to report for the services secondary to work or other personal commitments. Our recommendation was earlier suggested in China as a measure of improving linkage to care among men [53, 62]. By extending hours of operation, a centre would be customizing HIV services to the needs of a client [63]. Additionally, this finding builds on a recommendation from earlier studies that reported that HIV-infected participants preferred longer opening hours which will lessen the waiting time at a facility [62, 64]. Currently in Malawi, this measure has only been applied to HIV Testing services, and going forward, it would be important to evaluate this strategy with ART clinics [33].

The creation of male-only-clinics as reported in our study builds on literature that has reported that the creation of male-only-avenues for HIV testing without corresponding ART services on the spot results in the non-initiation of ARVs [65]. Nonetheless, programmatic reports suggest that male-only clinics and extension of hours for them to initiate ARTs were effective in reaching more men with services [65]. Having male-only clinics extends the notion of men supporting each other in peer groups which diffuse masculinity ideologies that affect engagement with HIV services [66]. These male-friendly spaces will also diffuse the notions of viewing oneself as "the real man" resilient to illnesses which inadvertently impedes ART initiation [19]. Programmatic data on the implementation of male-only clinics in Malawi show that they are effective with the potential for scaling [33]. As Malawi advances in the provision of ART services, the incorporation of adherence clubs as ART initiation avenues could be explored [67].

Offering more drugs as suggested in this study is a strategy that was already implemented but requires strengthening to yield more positive outcomes [59, 68]. Previous studies have reported that offering stable client antiretrovirals twice a year is feasible and has resulted in better retention rates with a reduction in viral loads than those clients who were reviewed at the clinic every two months [69]. Furthermore, a Ugandan study reported that six-monthly refills of ARVs for stable clients minimized congestion [27]. Although Malawi implemented Multi-Month Scripting (MMS) for stable clients whereby clients are given ARVs to last them for six months, the effectiveness of these changes is yet to be assessed on a larger scale [56, 70].

Our study further advocates dispensing of ARVs from the main pharmacy and using the same outlet used for dispensing drugs for other illnesses because it will avert undue disclosure of an infected status. Interestingly, a study conducted in South Africa and Zambia showed that the South African participants were content with accessing their ARVs from the main pharmacy while their counterparts were unwilling to access their ARVs from the main pharmacy because they were subjected to long waiting times [60]. Arguably, using one pharmacy for all medications will require improving the number of health personnel and the system to ensure that patients have a quick exit. In Tanzania, patients on ARVs who shared the same waiting area with other patients but accessed their ARVS from a different outlet felt discriminated against and experienced unintended HIV status disclosure [49]. Our findings propose a solution by suggesting that services should not be segregated in the quest to offer quality services because it becomes a deterrent to the initiation and continued use of ARVs. Furthermore, fragmented service and assignment of clinics as per disease condition promotes stigmatization and unintended disclosure which impedes early initiation of ARVs [49]. Another measure to minimize unintended disclosure is through changing the packaging of ARVs to a more concealing one that does not unintendedly reveal one's status. This finding remains consistent with a Tanzanian study where people living with HIV would discard the ARV packaging to prevent

others from noticing that they are on ARVs [49]. This recommendation requires liaising with the funders of the ARVs for consideration.

Our study argues that counseling should emphasize the benefits of initiating ARVs early. This was also raised by a Chinese study whose participants complained of inadequate knowledge of HIV services and care which resulted in non-linkage in HIV care [53]. The information shared should include the preventive benefits that ARVs render, the advantages of the current regimes [71], and should embrace the changing landscape of HIV and AIDS guidelines [20]. Additionally, counseling should be tailored to the specific needs of a particular client and not general education. Previous studies have asserted that there is a need to increase awareness of HIV and AIDS issues among men to improve utilization [72].

The verbatim implementation of the test-and-treat strategy as stated in this study requires a change in the current operations of HIV services where men are tested and are scheduled to initiate ARVs on designated days. This mainly happens because the health personnel that conducts HIV tests are not certified as ART providers. The lack of properly integrated services requires clients to navigate several health care workers and physical areas in a health system which is also a barrier to the initiation of ARVs [73]. Malawi is yet to fully implement a verbatim test-and-treat approach and this is evidenced by the number of people that are not linked to care instantly which necessitates a review of health system operations if we are to eliminate factors that hinder implementation [74–76].

## Strengths and limitations

Although our study provides greater insights on what men prefer as strategies for initiating ART, caution has to be exercised in implementing our results since the nature of the design does not allow for generalizability. The use of only female data collectors may have limited men from opening up in sharing their views, however, the data collectors were trained on measures of ensuring that men were free to share their insights and also on probing for more information to achieve a comprehensive narrative. Our findings provide strategies that may be rolled out in research to assess various implementation outcomes. The strength of our study lies in the fact that it included men from different geographical areas and, various HIV statuses which broadened the scope of the response. The use of various methods of data collection from different populations is another strength because it provides comprehensive perspectives from varying stakeholders. Future research should focus on implementation strategies of dispensing ARVs daily and integration of the highlighted fragmented services within HIV care.

## Conclusion

As HIV and AIDS become a chronic disease, health systems need to be restructured to contain the condition into the routine systems with several pathways of accessing treatment that remain convenient and non-discriminatory to the users. Successful initiation and delivery of ARVs to men will require a revision of the service delivery framework that is currently in use in Malawi. Patient centeredness with a focus on the preference of men in accessing ART services is paramount and requires adapting HIV services to the needs of men in their engagement at every level of service delivery, design, and implementation. Elimination of all kinds of stigma especially those unintentionally imposed by the health system will improve the initiation of ARVs among men.

## Acknowledgments

We are grateful to all the study participants that participated in the study, the heads of each facility where we conducted the study and the Director for Health and Social Services for Blantyre in Malawi for institutional support.

## Author Contributions

**Conceptualization:** Alinane Linda Nyondo-Mipando.

**Data curation:** Alinane Linda Nyondo-Mipando, Leticia Suwedi Kapesa, Sangwani Salimu, Thokozani Kazuma.

**Formal analysis:** Alinane Linda Nyondo-Mipando, Leticia Suwedi Kapesa, Sangwani Salimu, Thokozani Kazuma.

**Funding acquisition:** Alinane Linda Nyondo-Mipando.

**Investigation:** Alinane Linda Nyondo-Mipando, Leticia Suwedi Kapesa, Sangwani Salimu, Thokozani Kazuma.

**Methodology:** Alinane Linda Nyondo-Mipando.

**Project administration:** Alinane Linda Nyondo-Mipando.

**Supervision:** Alinane Linda Nyondo-Mipando, Victor Mwapasa.

**Writing – original draft:** Alinane Linda Nyondo-Mipando.

**Writing – review & editing:** Alinane Linda Nyondo-Mipando, Leticia Suwedi Kapesa, Sangwani Salimu, Thokozani Kazuma, Victor Mwapasa.

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
