## [Decision Letter · Decision Letter 0]

31 Jul 2020

PONE-D-20-08357

“Dispense Antiretrovirals daily!” Restructuring delivery of HIV services to optimize antiretroviral initiation among men in Malawi.

PLOS ONE

Dear Dr. Nyondo-Mipando,

Thank you for submitting your manuscript to PLOS ONE. After careful consideration, we feel that it has merit but does not fully meet PLOS ONE’s publication criteria as it currently stands. Therefore, we invite you to submit a revised version of the manuscript that addresses the points raised during the review process.

We look forward to receiving your revised manuscript.

Kind regards,

Jeremiah Chikovore

Academic Editor

PLOS ONE

Journal Requirements:

Reviewers' comments:

Reviewer's Responses to Questions

**Comments to the Author**

1. Is the manuscript technically sound, and do the data support the conclusions?

Reviewer #1: Yes

Reviewer #2: Partly

2. Has the statistical analysis been performed appropriately and rigorously? 

Reviewer #1: N/A

Reviewer #2: N/A

3. Have the authors made all data underlying the findings in their manuscript fully available?

Reviewer #1: No

Reviewer #2: No

4. Is the manuscript presented in an intelligible fashion and written in standard English?

Reviewer #1: No

Reviewer #2: No

5. Review Comments to the Author

Reviewer #1: The manuscript addresses the important topic of barriers to ART initiation and adherence among men in Malawi. The manuscript is technically sound, but errors in the English and choppy sentence structure make comprehension difficult at times and interrupt the flow of the manuscript. The manuscript would benefit from heavy editing from a professional editor both in terms of English grammar and organization (flow from one sentence/topic to another; particularly in the introduction and discussion sections).

The methods section is well-developed according to COREQ guidelines.

Information about ethical approval should also be included in the body of the manuscript, in the methods section.

Line 240: Why were all data collectors female? How might that have affected data collection?

In the data analysis section, it would be interesting to provide examples of the thematic codes used.

Lines 489-495: Includes a quote that was included elsewhere.

Lines 508-515: It would be helpful to explain what the role of a guardian is in Malawi.

Given that men of unknown status, HIV-positive men on ARTs and HIV-positive men not on ARTs were sampled, it would be interesting to note whether there were differences in the insights obtained from each category of men.

Given that the study was conducted in 2017, it would be helpful to explicitly discuss how relevant the findings are in 2020. Has Malawi already rolled-out policy changes that address any of the study's recommendations? If so, is there any evidence that those policy changes have improved uptake of timely and continuous ART by men in Malawi?

Regarding data availability, there are many participant quotes in the manuscript, but the authors could also make the de-identified transcripts or excerpts of transcripts reflecting the thematic codes underlying the manuscript available on a public data repository.

Reviewer #2: The paper ‘Dispense Antiretrovirals daily!’ Restructuring delivery of HIV services to optimize ART initiation among men in Malawi is a qualitative study that evaluates the existing health services for men in Blantyre, Malawi. This paper fits into the small but important body of literature from SSA that focuses on HIV service provision for men, a relatively under-researched topic within the HIV literature.

This qualitative paper is containing findings from an extensive study for which the research team interviewed a range of different health workers and male patients recruited from 7 different clinics in the city. I applaud the authors for being able to get so many rich findings, but unfortunately, I have to reject the paper. Below, I will outline the reasons why I made this decision and will give some ideas on how to move forward.

Firstly, the structure of the background and the rest of the paper is not consistent. In the background, a brief overview of the problem at hand is given, but there is little description on how the services are structured within the area, what exact services are offered per clinic and how these clinics work organisationally. This is key for the rest of the paper. Ideas around stigma and other barriers to care for men are mentioned, but not adequately unpacked, even though many of the findings point towards stigmatization of men when they come to the clinic. This is a missed opportunity.

Then, the methods is too long and too descriptive, and could be shorter. However, this can be done through thorough restructuring and proofreading of the paper.

In the results, I wanted to congratulate the authors with being able to recruit 15 male health workers, and I was looking forward to reading their remarks on male patient’s behaviours, but this was not provided in detail. Some of the findings section (1.1) were really long, and the quotes were not explained/contextualised within a structured narrative. Others were really short and provided little detail. There are several mentions of ‘a guardian’, even though I thought only men of 18 and above were recruited, which was confusing. A summary of the findings would have been useful and when reading the different sections, I would provide more details on how you interpreted the findings. I would also restructure the paragraphs and re-analyse part of the data, as some quotes are repetitive. I would definitely include a section on Stigma, and on Health System barriers.

In the discussion, little reference is made to existing strategies for men and little advise if given for implementing changes for future practice.

In summary, I think this paper could improve drastically if the authors worked with an independent writing advisor to restructure this paper, as I think the authors put in an incredible amount of work in getting the data and on this important topic. Overall editing is also encouraged to improve the paper.

6. PLOS authors have the option to publish the peer review history of their article (what does this mean?). If published, this will include your full peer review and any attached files.

Reviewer #1: No

Reviewer #2: **Yes: **Myrna van Pinxteren

---

## [Author Response · Author response to Decision Letter 0]

4 Sep 2020

Dear Editor and Reviewers,

PONE-D-20-08357: “Dispense Antiretrovirals daily!” Restructuring the delivery of HIV services to optimize antiretroviral initiation among men in Malawi.

We are thankful for the review over our manuscript which has not only strengthened the message but also enabled us to reflect on the data we had for a better output. The review was thorough and constructive. Please find below our responses to the queries as raised by the reviewers

Reviewer # 1

1. The manuscript addresses the important topic of barriers to ART initiation and adherence among men in Malawi. The manuscript is technically sound, but errors in the English and choppy sentence structure make comprehension difficult at times and interrupt the flow of the manuscript. The manuscript would benefit from heavy editing from a professional editor both in terms of English grammar and organization (flow from one sentence/topic to another; particularly in the introduction and discussion sections).

Response: The manuscript has been edited for language and we have revised our paragraphs to highlight topic sentences thereby limiting one idea or only related ideas under one paragraph to promote comprehension. This is reflected mainly in the introduction and discussion sections of the manuscript.

2. The methods section is well-developed according to COREQ guidelines.

Response: We appreciate this comment.

3. Information about ethical approval should also be included in the body of the manuscript, in the methods section.

Response: We have included the ethics statement in the main body as reflected in lines 289-309:

Ethical Considerations

Our study obtained ethical approval from Malawi’s College of Medicine Research and Ethics Committee (COMREC- Number P.11/16/2064) prior to study implementation. The District of Health and Social Services (DHSS) for Blantyre and the Director of (Queen Elizabeth Central Hospital (QECH) provided Institutional support for the study to take place in the respective sites. Each study participant provided a written informed consent prior to any study procedures. Participants that could not read nor write thumb-printed on the consent form after it was read to them in the presence of an impartial witness. We assured all participants that their participation in the study would not affect their receipt of medical services at the respective facilities. As a measure of maintaining participants’ confidentiality, anonymity and privacy, we conducted all interviews in a private and quiet room at the facility as per preference of the interviewees. The HIV and ART Coordinators’ interviews were conducted at their office stations as preferred by them. The FGDs were all conducted within a health facility however HIV statuses were not disclosed during the discussion to maintain privacy. The men with an unknown HIV status were referred for HIV testing while those who were HIV infected and not on medications yet were referred for ART as per national guidelines. Interviews with HCWs were conducted after working hours, during lunch breaks and weekends as per participant’s preference to avoid disruption of work. We used codes instead of participants’ names in the study summaries. All audios and transcripts were saved in a password protected computer with limited access to the researchers

4. Line 240: Why were all data collectors female? How might that have affected data collection?

Response: We realise that having only female data collectors may bias our findings, we have acknowledged it as a limitation for the study. This is stated as follows on page:

Lines 802-806: The use of only female data collectors may have limited men from opening up in sharing their views, however, the data collectors were trained on measures of ensuring that men were free to share their insights and also on probing for more information to achieve a comprehensive narrative. 

5. In the data analysis section, it would be interesting to provide examples of the thematic codes used.

Response: We have included some examples that led us to realise the thematic codes we presented. See Lines 274-286:

Our three main themes were realised through the collation of similar, related, and recurrent codes, specifically, we grouped all codes on drugs such as dispensation, composition, packaging, and innovations under one overarching theme of differentiating drug dispensing modalities; all codes related to clinic flow, layout and hours and days of operation of a facility were aggregated under Differentiated clinic operations and all codes that referred to the implementation of test and treat and counseling were grouped under the strengthening implementation of available policies theme. We examined each code for further subcategories (40) to ensure that each related concept is grouped under one category and we dropped themes that were not broad enough, for instance, we had a theme on male-specific clinics which we later grouped under clinic operations because it was closely related to that. Fourthly, the research team reviewed the themes and we refined them as appropriate to achieve a correct representation of our findings [

6. Lines 489-495: Includes a quote that was included elsewhere.

Response: We have deleted the repeated quote and have revised that section.

7. Lines 508-515: It would be helpful to explain what the role of a guardian is in Malawi.

Response: We have included information about a guardian and their role in the Malawian setting in the background to our study and it reads as follows:

Lines 112-116: At the point of ART initiation, a client is supposed to nominate a person who is commonly referred to as a “guardian” who is usually a layperson and could be a friend or family member (32). A guardian is counseled along with a client on the HIV status of the client and information on ARVS to promote adherence and at times to collect the ARVs on behalf of the client

8. Given that men of unknown status, HIV-positive men on ARTs and HIV-positive men not on ARTs were sampled, it would be interesting to note whether there were differences in the insights obtained from each category of men.

Response: We reviewed our data set to ensure that we have correctly represented the views and have highlighted in the narratives areas where various men of different HIV statuses shared their views. These are reflected in lines 362, 377, 427 and 472. Our results should also be considered under the context that men with an unknown HIV status were limited in their comments over other parts of ART services as they have not interfaced with them under their status.

9. Given that the study was conducted in 2017, it would be helpful to explicitly discuss how relevant the findings are in 2020. Has Malawi already rolled-out policy changes that address any of the study's recommendations? If so, is there any evidence that those policy changes have improved uptake of timely and continuous ART by men in Malawi?

Response: We have included this information on the changes that have occurred since then and this is reflected as follows:

Lines 116-124: Despite the lag in men accessing HIV services, there have been no specific HIV service programmes rolled out for men in Malawi. However, anecdotal reports show that some health implementing partners piloted male-friendly clinic services in 12 primary health facilities in Blantyre, Malawi in 2018, a year after we had finalized data collection. The male-friendly clinics would open once a week on Saturday’s targeting newly diagnosed men and those that had problems collecting ARVs during the week. Male- friendly clinics were run by both female and male health care workers providing non-communicable health services, sexually transmitted infection services, HIV testing services, and ARVs dispensation.

Lines 716-718: A similar measure of extending hours of operation in HIV services in Malawi was only applied to HIV Testing services and going forward, it would be important to evaluate this strategy with ART clinics.

Lines 700- 703: Nonetheless, programmatic reports suggest that men advocated for male-only clinics and extensions of hours for them to initiate ARTs which has been tested on a small scale in Malawi and may benefit from further research.

Lines 721-724: Programmatic data on the implementation of male-only clinics in Malawi show that they are effective with the potential for scaling (63).

Lines 740-745: Specifically for Malawi, Multi-Month Scripting (MMS) for stable clients was implemented and clients are given ARVs to last them for six months The goal of MMS is to reduce the number of visits to the clinics for refills which also creates a platform for health facilities to focus more on complex clients that need more attention compared to stable ones. The effectiveness of these changes is yet to be assessed on a larger scale in Malawi (51,66).

Lines 788-791: Malawi is yet to fully implement a verbatim of test and treat approach and this is evidenced by the number of people that are not linked to care instantly which necessitates a review of health system operations to eliminate factors that hinder implementation (70–72). 

10. Regarding data availability, there are many participant quotes in the manuscript, but the authors could also make the de-identified transcripts or excerpts of transcripts reflecting the thematic codes underlying the manuscript available on a public data repository.

Response: Currently, the data are still being used in manuscript writing by the authors and students within College of Medicine, After all those commitments, the data will be made available in an open domain. However, the data sets can be made available at reasonable requests from the corresponding author in compliance with College of Medicine’s requirements. 

Reviewer # 2

1. The paper ‘Dispense Antiretrovirals daily!’ Restructuring delivery of HIV services to optimize ART initiation among men in Malawi is a qualitative study that evaluates the existing health services for men in Blantyre, Malawi. This paper fits into the small but important body of literature from SSA that focuses on HIV service provision for men, a relatively under-researched topic within the HIV literature.

This qualitative paper is containing findings from an extensive study for which the research team interviewed a range of different health workers and male patients recruited from 7 different clinics in the city. I applaud the authors for being able to get so many rich findings, but unfortunately, I have to reject the paper. Below, I will outline the reasons why I made this decision and will give some ideas on how to move forward.

Response: We appreciate your taking the time to review the paper and have found your comments and suggestions valuable.

2. Firstly, the structure of the background and the rest of the paper is not consistent. In the background, a brief overview of the problem at hand is given, but there is a little description of how the services are structured within the area, what exact services are offered per clinic and how these clinics work organisationally. This is key for the rest of the paper. 

Response: We have included a section that discusses on how services are provided and organized in Malawi and agree with the reviewer that this component is key to the paper. The following has been included, see Lines 104 to 124

HIV and ART services in Malawi are provided for free in all Government-owned health facilities and at a cost in private and faith-based organisations. The services include HIV testing, ART initiation, and follow-up in care. Health facilities in the urban areas offer HIV services daily while those in semi-urban and rural areas have a designated day within the week when ART clinics run while HIV testing is offered daily. Malawi has integrated the delivery of HIV services with Tuberculosis, Family planning, and Sexual and reproductive health programmes (30). All services are standardized and follow the same monitoring plan regardless of their location and rarely do facilities experience stock-outs of ARVs (31). At the point of ART initiation, a client is supposed to nominate a person who is commonly referred to as a “guardian” who is usually a layperson and could be a friend or family member (32). A guardian is counseled along with a client on the HIV status of the client and information on ARVS to promote adherence and at times to collect the ARVs on behalf of the client. Despite the lag in men accessing HIV services, there have been no specific HIV service programmes rolled out for men in Malawi. However, anecdotal reports show that some health implementing partners piloted male-friendly clinic services in 12 primary health facilities in Blantyre, Malawi in 2018, a year after we had finalized data collection. The male-friendly clinics would open once a week on Saturday’s targeting newly diagnosed men and those that had problems collecting ARVs during the week. Male- friendly clinics were run by both female and male health care workers providing non-communicable health services, sexually transmitted infection services, HIV testing services, and ARVs dispensation. 

3. Ideas around stigma and other barriers to care for men are mentioned, but not adequately unpacked, even though many of the findings point towards stigmatization of men when they come to the clinic. This is a missed opportunity.

Response: We have provided more depth in the narratives of our quotes to shed more on stigmatization and the rationale for the suggested strategies. This is reflected in the results and discussion sections on the following lines: 363-368, 375-380, 388-392, 444-447, 472-479, 572-576, 585-592, 668-675, 676-685, 703-708, 746-751, and 756-766.

4. Then, the methods is too long and too descriptive, and could be shorter. However, this can be done through thorough restructuring and proofreading of the paper.

Response: We believe that our methods section is lengthy mainly because we applied the COREQ guidelines to ensure that it retains the quality as needed for a qualitative paper. This paper is also the first one from our project on HIV and men as such we also wanted it to have the methods laid out in detail so that future papers would refer this paper. 

5. In the results, I wanted to congratulate the authors with being able to recruit 15 male health workers, and I was looking forward to reading their remarks on male patient’s behaviours, but this was not provided in detail. Some of the findings section (1.1) were really long, and the quotes were not explained/contextualised within a structured narrative. Others were really short and provided little detail. There are several mentions of ‘a guardian’, even though I thought only men of 18 and above were recruited, which was confusing.

Response: We acknowledge that the minimal clarification on a guardian was confusing and we have included a section in the introduction that clarifies who a guardian is to better contextualize our reference on the same. This is reflected as follows:

Lines 112-116: At the point of ART initiation, a client is supposed to nominate a person who is commonly referred to as a “guardian” who is usually a layperson and could be a friend or family member (32). A guardian is counseled along with a client on the HIV status of the client and information on ARVS to promote adherence and at times to collect the ARVs on behalf of the client

6. A summary of the findings would have been useful and when reading the different sections, I would provide more details on how you interpreted the findings. I would also restructure the paragraphs and re-analyse part of the data, as some quotes are repetitive. I would definitely include a section on Stigma, and on Health System barriers.

Response: We have added explanations regarding the suggestions and have alluded to stigma and health system barriers that may have influenced the participants to suggest the stated strategies. We have however not included a section that specifies barriers as that would be outside the objective of this paper.

We have removed repetitive quotes and have grouped similar ideas under one narrative. The aspects of stigma are alluded as in response to Query Number 3, Reviewer 2

7. In the discussion, little reference is made to existing strategies for men and little advise if given for implementing changes for future practice.

Response: We have included the following statements in the introduction and discussion sections to highlight the existing strategies and advice:

Lines 116-124: Despite the lag in men accessing HIV services, there have been no specific HIV service programmes rolled out for men in Malawi. However, anecdotal reports show that some health implementing partners piloted male-friendly clinic services in 12 primary health facilities in Blantyre, Malawi in 2018, a year after we had finalized data collection. The male-friendly clinics would open once a week on Saturday’s targeting newly diagnosed men and those that had problems collecting ARVs during the week. Male- friendly clinics were run by both female and male health care workers providing non-communicable health services, sexually transmitted infection services, HIV testing services, and ARVs dispensation.

Lines 716-718: A similar measure of extending hours of operation in HIV services in Malawi was only applied to HIV Testing services and going forward, it would be important to evaluate this strategy with ART clinics.

Lines 700- 703: Nonetheless, programmatic reports suggest that men advocated for male-only clinics and extensions of hours for them to initiate ARTs which has been tested on a small scale in Malawi and may benefit from further research.

Lines 721-724: Programmatic data on the implementation of male-only clinics in Malawi show that they are effective with the potential for scaling (63).

Lines 740-745: Specifically for Malawi, Multi-Month Scripting (MMS) for stable clients was implemented and clients are given ARVs to last them for six months The goal of MMS is to reduce the number of visits to the clinics for refills which also creates a platform for health facilities to focus more on complex clients that need more attention compared to stable ones. The effectiveness of these changes is yet to be assessed on a larger scale in Malawi (51,66).

Lines 788-791: Malawi is yet to fully implement a verbatim of test and treat approach and this is evidenced by the number of people that are not linked to care instantly which necessitates a review of health system operations to eliminate factors that hinder implementation (70–72). 

Lines 716-718: A similar measure of extending hours of operation in HIV services in Malawi was only applied to HIV Testing services and going forward, it would be important to evaluate this strategy with ART clinics.

Lines 700- 703: Nonetheless, programmatic reports suggest that men advocated for male-only clinics and extensions of hours for them to initiate ARTs which has been tested on a small scale in Malawi and may benefit from further research.

Lines 721-724: Programmatic data on the implementation of male-only clinics in Malawi show that they are effective with the potential for scaling (63).

Lines 740-745: Specifically for Malawi, Multi-Month Scripting (MMS) for stable clients was implemented and clients are given ARVs to last them for six months The goal of MMS is to reduce the number of visits to the clinics for refills which also creates a platform for health facilities to focus more on complex clients that need more attention compared to stable ones. The effectiveness of these changes is yet to be assessed on a larger scale in Malawi (51,66).

Lines 788-791: Malawi is yet to fully implement a verbatim of test and treat approach and this is evidenced by the number of people that are not linked to care instantly which necessitates a review of health system operations to eliminate factors that hinder implementation (70–72). 

8. In summary, I think this paper could improve drastically if the authors worked with an independent writing advisor to restructure this paper, as I think the authors put in an incredible amount of work in getting the data and on this important topic. Overall editing is also encouraged to improve the paper.

Response: The paper has been revised and edited accordingly

Sincerely,

Alinane Linda Nyondo-Mipando RNM, Ph.D (Corresponding Author)

---

## [Decision Letter · Decision Letter 1]

27 Oct 2020

PONE-D-20-08357R1

“Dispense Antiretrovirals daily!” Restructuring the delivery of HIV services to optimize antiretroviral initiation among men in Malawi.

PLOS ONE

Dear Dr. Nyondo-Mipando,

Thank you for submitting your manuscript to PLOS ONE. After careful consideration, we feel that it has merit but does not fully meet PLOS ONE’s publication criteria as it currently stands. Therefore, we invite you to submit a revised version of the manuscript that addresses the points raised during the review process.

We look forward to receiving your revised manuscript.

Kind regards,

Jeremiah Chikovore

Academic Editor

PLOS ONE

Additional Editor Comments (if provided):

Thank you for submitting a revised version of your manuscript. The feedback from reviewers is that, overall the article has improved. However, we cannot accept the article in its current form. Please address all of the reviewers’ concerns fully.

A message that comes through from the reviews is that the article still needs to be edited for flow, grammar, consistency, and redundancies. Whereas one of our reviewers has considered helping with editing, the journal determines rather that this editing be pursued independently by yourselves as authors. Kindy note that PLOS partners with Editage for purposes of editing and would be happy to connect you to this service, while you are also free to use whichever resources you prefer.

Reviewers' comments:

Reviewer's Responses to Questions

**Comments to the Author**

1. If the authors have adequately addressed your comments raised in a previous round of review and you feel that this manuscript is now acceptable for publication, you may indicate that here to bypass the “Comments to the Author” section, enter your conflict of interest statement in the “Confidential to Editor” section, and submit your "Accept" recommendation.

Reviewer #1: (No Response)

Reviewer #2: (No Response)

2. Is the manuscript technically sound, and do the data support the conclusions?

Reviewer #1: Yes

Reviewer #2: Yes

3. Has the statistical analysis been performed appropriately and rigorously? 

Reviewer #1: N/A

Reviewer #2: Yes

4. Have the authors made all data underlying the findings in their manuscript fully available?

Reviewer #1: No

Reviewer #2: Yes

5. Is the manuscript presented in an intelligible fashion and written in standard English?

Reviewer #1: No

Reviewer #2: No

6. Review Comments to the Author

Reviewer #1: Thank you for the revision. The manuscript is much improved, but it still requires a careful edit to improve the English and the flow, facilitate comprehension and to make it as succinct as possible/eliminate redundancies. With substantial editing, I believe it would be fit to publish as the methodology is solid and the results interesting.

I began making suggested edits, but it became too time consuming to continue in this way:

Lines 89-91: There are two ideas presented in this sentence, but I'm not sure that one follows from the other: The services are also biased towards females (21) such that projects among men have primarily focused on men having sex with men (MSM) (22,23).

Line 115: ARVs instead of ARVS

Line 150: Delete the second "rate": Blantyre district has the highest prevalence rate of HIV among men at 14% as of...

Line 154: IDIs instead of IDIS

Lines 155-157: The following sentence is vague; consider rephrasing or deleting: We deliberately sampled men of varying HIV statuses to gather perceptions of all characteristics of men to ensure wider coverage of our findings.

Line 176: Consider changing "after sharing the study and its eligibility, the facility In-charge" to "after explaining the study and the eligibility criteria, the Officer In-charge assisted with the identification of..."

Lines 186-191: I would rephrase as: "We included heterosexual men (as ascertained by asking them their

sexual orientation) who were 18 years of age and above, able and willing to provide consent, and of a range of HIV statuses: uninfected (confirmed through checking the health passport books), unknown (ascertained by asking if they have ever tested for HIV or not), and HIV infected on ART and not on ART (confirmed through checking the health passport book and/or clinic records)."

Line 191: There are places where you are using "facility In-charge" where I think you may mean "Officer In-charge"?

Line 207: "assistance from HCWs" instead of "assistance for"

Lines 208-211: Split into two sentences: We included men that were 18 years and above, varying HIV status, on ARVs

and not on ARVs, able to provide consent, and willing to participate in the study. These men were identified in the departments within the clinic and the communities around each health facility.

Lines 211-212: I would rephrase as: "Men were scheduled for a discussion at a time and place convenient for them."

Lines 219-220: I would rephrase as: With the exception of VM, all authors conducted interviews and all data collectors were female."

Line 220: I would add: "Before data collection, ALNM, the Principal Investigator (PI) trained..."

Line 221-225: I would rephrase as: "LSK and SS are public health specialists with Masters in Public health. TK has a Bachelor of Nursing and Midwifery and ALNM has a PH.D. in Health Systems and Policy."

Line 230: Replace "shared of her past research" with "shared her past research"

Line 234: add an "s" to question

Lines 235-238: Were these the broad questions for everyone or just for health care workers? Please clarify.

Lines 246-250: This sentence is a bit choppy and unclear.

Line 251: Replace "in a language" with "in the language"

Line 260: NVivo instead of NVIvo

Lines 263-265: Suggest rephrasing as: Codes were generated, deductively, from the objectives of the study and also, inductively, from the data.

Lines 265-267: Suggest editing to: "First, the Principal Investigator listened to the audios and read the transcripts multiple times to gain a deeper understanding of the issues raised"

Table 1: Overall, the table is a bit confusing and needs improvement. I suggest adding a column for percentages as well as for counts. In terms of education level, those who did not have a primary school education had no education? Or more than a primary school education? Perhaps it would be clearer to show each of the different levels of education represented. Uptake of ARVs is only relevant for those who are HIV-positive, not the full sample of 131 participants, so it is difficult to interpret.

Table 2: Make it clear that the table summarizes the key findings.

The discussion section and conclusion require editing and focus to improve comprehension and retention of the key contributions of the manuscript.

Reviewer #2: Comments for the authors:

Thank you so much for giving me the opportunity to review this paper. The structure and writing of this paper has been drastically improved compared to the previous draft. However, I still am of the opinion that the methods section and section 1.1 in the findings is too detailed and can be shortened for readability in the paper. I have also given some suggestions for further readings for the discussion section. Lastly, I would urge the authors to get a professional editor to proofread the publication for grammar, spelling and overall readability of the paper. Please see more detailed comments below. These comments can be addressed with the assistance of the editor.

Introduction:

For the following statements, there is there no reference, not even a webpage, health report or policy brief. ‘The implementation of these services must be based on evidence and this is important for your paper However, anecdotal reports show that some health implementing partners piloted male-friendly clinic services in 12 primary health facilities in Blantyre, Malawi in 2018, a year after we had finalized data collection.’ The implementation of these services must be based on evidence and this is important for your paper.

Methods:

I am still of the opinion that this section is too long and detailed. Although your comment suggest that this is the first publication from the project, it is not a description of the project or M&E paper, therefore there is no need to describe the data collection processes in too much detail. A few more points below which can help you to condense the method section:

a) References on in-depth interviews and other data collection methods are missing (line 131). Identification of participants (both patients and health care workers) for IDI’s and FGD’s could be summarized further. You are loosing the reader here.

b) Data analysis can be further summarized too, the process is interesting, but written to extensively. take out for instance the following sentence: ‘We examined each code for further subcategories (40) to ensure that each related concept is grouped under one category and we dropped themes that were not broad enough, for instance, we had a theme on male-specific clinics which we later grouped under clinic operations because it was closely related to that.’ (line 282 – 284)

Findings:

I still believe that section one of the findings can be shortened, to improve the flow and structure of the article. Paragraph 1.1 contains too many quotes which makes it too repetitive. Pick a few good ones to make your argument and move on to the next paragraph.

Also, be very clear about the description of participants and be consistent. The first quote starting on line 365, states ‘younger men FGD participant at Centre 2’. This sentence is incorrect, unless you are quoting 2 men. I would change it to; ‘Male participant FGD at Centre 2’. This is done throughout the article. Also, is it relevant to know if the participant is infected with HIV or not? And are they honest about it? I would take that out of the descriptions of the quotes as done in line 380.

At line 390 of the findings, again the description of the health care worker is different, here you say ‘KII, HDA at Centre 3’, please be specific or explain the acronyms at the start of the findings. This is about consistency.

In Paragraph 1.2, you want to make a clearer connection between the different paragraphs and signpost the reader. An example from line 450, you can add; ‘in addition to dispensing ARV’s on a daily basis, participants also suggested longer opening times for clinics to accommodate work schedules of men.’ The same goes for other paragraphs in the findings section.

Although this part needs editing and further proofreading, the analysis of the findings and unpacking of the quotes has definitely improved throughout the findings.

Discussion:

Stigmatization and HIV services has been unpacked accurately in the discussion section. There is a missing reference after the sentence in line 722.

The paragraph starting on line 736 gives a good indication of why ARV’s need to be dispensed not per month, but per 3 or 6 months. But there is a need to include that this only works with patients who are stable on ARV’s. In South Africa, stable means being on treatment uninterrupted for at least six months. Another angle to explore can be pill-fatigue, a concept that has come up in similar conversations we had in our research project. Providing treatment for 6 months can prevent pill-fatigue among patients which can lead to non-adherence.

The paragraph starting on line 747 speaks about dispensary of ARV’s in other places than a pharmacy. Here, you can link effectively to studies conducted in South Africa with adherence clubs that meet patients outside of clinic spaces. Medication gets dispensed in community centres or churches, which means patients don’t have to wait in queues, have more freedom and keep their confidentiality. I think referring to some of these studies will improve the discussion section of the paper.

Conclusion:

I would suggest not to start with a statement about HIV stigma and services for men, as this was not the focus of your paper. The focus of your paper is how health services can be better tailored for the needs of men.

7. PLOS authors have the option to publish the peer review history of their article (what does this mean?). If published, this will include your full peer review and any attached files.

Reviewer #1: No

Reviewer #2: **Yes: **Myrna van Pinxteren

---

## [Author Response · Author response to Decision Letter 1]

2 Dec 2020

Dear Editor and Reviewers,

PONE-D-20-0857R1- “Dispense Antiretrovirals daily!” Restructuring the delivery of HIV services to optimize antiretroviral initiation among men in Malawi.

We sincerely thank you for taking the time to review our revised manuscript. We have addressed the queries raised and find below our responses.

1. Reviewer #1: Thank you for the revision. The manuscript is much improved, but it still requires a careful edit to improve the English and the flow, facilitate comprehension and to make it as succinct as possible/eliminate redundancies. With substantial editing, I believe it would be fit to publish as the methodology is solid and the results interesting.

Response: We have had the article edited for language

2. I began making suggested edits, but it became too time consuming to continue in this way:

Lines 89-91: There are two ideas presented in this sentence, but I'm not sure that one follows from the other: The services are also biased towards females (21) such that projects among men have primarily focused on men having sex with men (MSM) (22,23).

Response: We have revised the sentence and it is in two parts now as follows: The services are also biased towards females (21) which has resulted in limited services for men. The services that are available for men have primarily focused on men having sex with men (MSM) and not the general population of men(22,23). Refer to page 4.

3. Line 115: ARVs instead of ARVS

Response: This has been updated

4. Line 150: Delete the second "rate": Blantyre district has the highest prevalence rate of HIV among men at 14% as of...

Response: The second “rate” has been deleted.

5. Line 154: IDIs instead of IDIS

Response: This has been corrected

6. Lines 155-157: The following sentence is vague; consider rephrasing or deleting: We deliberately sampled men of varying HIV statuses to gather perceptions of all characteristics of men to ensure wider coverage of our findings.

Response: We have deleted the sentence.

7. Line 176: Consider changing "after sharing the study and its eligibility, the facility In-charge" to "after explaining the study and the eligibility criteria, the Officer In-charge assisted with the identification of..."

Response: We have revised the sentence as suggested n line 165

8. Lines 186-191: I would rephrase as: "We included heterosexual men (as ascertained by asking them their

sexual orientation) who were 18 years of age and above, able and willing to provide consent, and of a range of HIV statuses: uninfected (confirmed through checking the health passport books), unknown (ascertained by asking if they have ever tested for HIV or not), and HIV infected on ART and not on ART (confirmed through checking the health passport book and/or clinic records)."

Response: We have revised the sentence as suggested. Refer to lines 175-179.

9. Line 191: There are places where you are using "facility In-charge" where I think you may mean "Officer In-charge"?

Response: We have made this consistent by using one term throughout.

10. Line 207: "assistance from HCWs" instead of "assistance for"

Response: We have changed it as suggested. 

11. Lines 208-211: Split into two sentences: We included men that were 18 years and above, varying HIV status, on ARVs and not on ARVs, able to provide consent, and willing to participate in the study. These men were identified in the departments within the clinic and the communities around each health facility.

Response: The sentence has been split into 2 as suggested. Refer to lines 191-193.

12. Lines 211-212: I would rephrase as: "Men were scheduled for a discussion at a time and place convenient for them."

Response: The sentence has been rephrased. Refer to lines 192-193.

13. Lines 219-220: I would rephrase as: With the exception of VM, all authors conducted interviews and all data collectors were female."

Response: This has been updated as suggested. Refer to lines 199-200.

14. Line 220: I would add: "Before data collection, ALNM, the Principal Investigator (PI) trained..."

Response: This has been updated as suggested. Refer to line 200.

15. Line 221-225: I would rephrase as: "LSK and SS are public health specialists with Masters in Public health. TK has a Bachelor of Nursing and Midwifery and ALNM has a PH.D. in Health Systems and Policy."

Response: This has been updated as suggested. Refer to lines 202-204

16. Line 230: Replace "shared of her past research" with "shared her past research"

Response: The word “of” has been deleted.

17. Line 234: add an "s" to question

Response: We have added an “s” to question. Refer to line 213.

18. Lines 235-238: Were these the broad questions for everyone or just for health care workers? Please clarify.

Response: We have added that the broad questions were for both men and health care workers. To ease the understanding, we have added the interview and discussion guides as supplementary files.

19. Lines 246-250: This sentence is a bit choppy and unclear.

Response: We have revised the sentences and have separated all aspects pertaining to quality under one section. Refer to lines 225-239.

20. Line 251: Replace "in a language" with "in the language"

Response: We have revised as suggested. Refer to line 227.

21. Line 260: NVivo instead of NVIvo

Response: This has been corrected. Refer to line 242.

22. Lines 263-265: Suggest rephrasing as: Codes were generated, deductively, from the objectives of the study and also, inductively, from the data.

Response: We have revised as suggested. Refer to lines 245-246.

23. Lines 265-267: Suggest editing to: "First, the Principal Investigator listened to the audios and read the transcripts multiple times to gain a deeper understanding of the issues raised"

Response: We have revised as suggested. Refer to lines 246-248.

24. Table 1: Overall, the table is a bit confusing and needs improvement. I suggest adding a column for percentages as well as for counts. In terms of education level, those who did not have a primary school education had no education? Or more than a primary school education? Perhaps it would be clearer to show each of the different levels of education represented. Uptake of ARVs is only relevant for those who are HIV-positive, not the full sample of 131 participants, so it is difficult to interpret.

Response: We have revised the Table and have presented percentages as suggested although we are aware that is not consistent with reporting of qualitative research results. Refer to page 12.

25. Table 2: Make it clear that the table summarizes the key findings.

Response: This has been updated. Refer to line 304.

26. The discussion section and conclusion require editing and focus to improve comprehension and retention of the key contributions of the manuscript.

Response: We have edited the sections as requested. 

¬¬¬¬¬¬¬¬¬¬¬¬¬¬¬¬¬¬¬¬¬¬¬¬¬¬¬¬¬¬¬¬¬¬¬¬¬¬¬¬¬¬¬¬¬¬-________________________________________________________________________________

Reviewer #2: Comments for the authors:

1. Thank you so much for giving me the opportunity to review this paper. The structure and writing of this paper has been drastically improved compared to the previous draft. However, I still am of the opinion that the methods section and section 1.1 in the findings is too detailed and can be shortened for readability in the paper. I have also given some suggestions for further readings for the discussion section. Lastly, I would urge the authors to get a professional editor to proofread the publication for grammar, spelling and overall readability of the paper. Please see more detailed comments below. These comments can be addressed with the assistance of the editor.

Response: Thank you for reviewing our manuscript once again. We have shortened section 1.1 and have had the manuscript edited for grammar, spelling and overall readability of the paper.

2. Introduction:

For the following statements, there is there no reference, not even a webpage, health report or policy brief. ‘The implementation of these services must be based on evidence and this is important for your paper However, anecdotal reports show that some health implementing partners piloted male-friendly clinic services in 12 primary health facilities in Blantyre, Malawi in 2018, a year after we had finalized data collection.’ The implementation of these services must be based on evidence and this is important for your paper. 

Response: We have provided the reference, reference number 33

3. Methods:

I am still of the opinion that this section is too long and detailed. Although your comment suggest that this is the first publication from the project, it is not a description of the project or M&E paper, therefore there is no need to describe the data collection processes in too much detail. 

Response: We have shortened the methods section as suggested.

A few more points below which can help you to condense the method section:

4. References on in-depth interviews and other data collection methods are missing (line 131). Identification of participants (both patients and health care workers) for IDI’s and FGD’s could be summarized further. You are loosing the reader here.

Response: We have summarized these further and have provided the omitted references. The references are inserted in line 132.

5. Data analysis can be further summarized too, the process is interesting, but written to extensively. take out for instance the following sentence: ‘We examined each code for further subcategories (40) to ensure that each related concept is grouped under one category and we dropped themes that were not broad enough, for instance, we had a theme on male-specific clinics which we later grouped under clinic operations because it was closely related to that.’ (line 282 – 284)

Response: We have shortened the data analysis section however we have retained some aspects that we were asked to include after the initial review.

6. Findings:

I still believe that section one of the findings can be shortened, to improve the flow and structure of the article. Paragraph 1.1 contains too many quotes which makes it too repetitive. Pick a few good ones to make your argument and move on to the next paragraph.

Response: We have retained those that are quite pertinent to the study. 

7. Also, be very clear about the description of participants and be consistent. The first quote starting on line 365, states ‘younger men FGD participant at Centre 2’. This sentence is incorrect, unless you are quoting 2 men. I would change it to; ‘Male participant FGD at Centre 2’. This is done throughout the article. 

Response: We have retained Younger Men because our FGDs were divided into older and younger men.

8. Also, is it relevant to know if the participant is infected with HIV or not? And are they honest about it? I would take that out of the descriptions of the quotes as done in line 380.

Response: This kind of identification was brought it after round 1 review. We now need further direction whether this has to be dropped or left as it is. 

9. At line 390 of the findings, again the description of the health care worker is different, here you say ‘KII, HDA at Centre 3’, please be specific or explain the acronyms at the start of the findings. This is about consistency.

Response: Please note that all abbreviations have been stated in full at the beginning and we have revised presentation of identifiers to remain consistent.

10. In Paragraph 1.2, you want to make a clearer connection between the different paragraphs and signpost the reader. An example from line 450, you can add; ‘in addition to dispensing ARV’s on a daily basis, participants also suggested longer opening times for clinics to accommodate work schedules of men.’ The same goes for other paragraphs in the findings section.

Although this part needs editing and further proofreading, the analysis of the findings and unpacking of the quotes has definitely improved throughout the findings.

Response: We have had to shorten the quotes used and that part has been take out. We have made connection statements where applicable in the results and discussion section of our manuscript.

11. Discussion:

Stigmatization and HIV services has been unpacked accurately in the discussion section. There is a missing reference after the sentence in line 722.

Response: We appreciate this comment.

12. The paragraph starting on line 736 gives a good indication of why ARV’s need to be dispensed not per month, but per 3 or 6 months. But there is a need to include that this only works with patients who are stable on ARV’s. In South Africa, stable means being on treatment uninterrupted for at least six months. Another angle to explore can be pill-fatigue, a concept that has come up in similar conversations we had in our research project. Providing treatment for 6 months can prevent pill-fatigue among patients which can lead to non-adherence.

Response: This was already presented on line 520. We have taken note of pill – fatigue and this has been recommended as part of future studies, considering that if we start on it in the discussion, we may digress from the core objective of the study.

13. The paragraph starting on line 747 speaks about dispensary of ARV’s in other places than a pharmacy. Here, you can link effectively to studies conducted in South Africa with adherence clubs that meet patients outside of clinic spaces. Medication gets dispensed in community centres or churches, which means patients don’t have to wait in queues, have more freedom and keep their confidentiality. I think referring to some of these studies will improve the discussion section of the paper.

Reference- We have made reference to a review on Adherence clubs. Refer to line 561.

14. Conclusion:

I would suggest not to start with a statement about HIV stigma and services for men, as this was not the focus of your paper. The focus of your paper is how health services can be better tailored for the needs of men.

Response: We have revised the conclusion and it reads as follows (Lines 623-631):

As HIV and AIDS become a chronic disease, health systems need to be restructured to contain the condition into the routine systems with several pathways of accessing treatment that remain convenient and non-discriminatory to the users. Successful initiation and delivery of ARVs to men will require a revision of the service delivery framework that is currently in use in Malawi. Patient centeredness with a focus on the preference of men in accessing ART services is paramount and requires adapting HIV services to the needs of men in their engagement at every level of service delivery, design, and implementation. Elimination of all kinds of stigma especially those unintentionally imposed by the health system will improve the initiation of ARVs among men. 

We remain grateful for the constructive review of our manuscript.

Sincerely,

Alinane Linda Nyondo-Mipando, RNM, Ph.D. (Corresponding Author)

---

## [Editor Report · Decision Letter 2]

5 Jan 2021

PONE-D-20-08357R2

“Dispense Antiretrovirals daily!” Restructuring the delivery of HIV services to optimize antiretroviral initiation among men in Malawi.

PLOS ONE

Dear Dr. Nyondo-Mipando,

Thank you for submitting your manuscript to PLOS ONE. After careful consideration, we feel that it has merit but does not fully meet PLOS ONE’s publication criteria as it currently stands. Therefore, we invite you to submit a revised version of the manuscript that addresses the points raised during the review process.

We look forward to receiving your revised manuscript.

Kind regards,

Jeremiah Chikovore

Academic Editor

PLOS ONE

Additional Editor Comments (if provided):

Please may you address the following additional queries.

- Line 33-35: Please remove capital letters in the phrases ‘in-depth interviews’, ‘focus group discussions’, ‘health care workers.’

- Line 36: Please remove ‘were’.

- Please check all places where ‘Initiation of antiretroviral drugs’ or similar is mentioned. I would presume it is ‘initiation of antiretroviral treatment’ rather than initiation of drugs.

- Please define ‘ART’, ‘ARVs’ at first use, including in abstract

- Line 39: Please check the sentence in abstract “Restructuring the delivery and conduct of ART clinics”. Do you intend to imply that ART clinics are delivered?

- Line 43-44. Consider restructuring/punctuating the sentence to enhance logic and ideas flow. It is not easy to understand specifically how ‘removing of other structural barrier’ links into the whole sentence.

- Line 45-46: ‘Implementation of test and treat strategy’. I think this phrase could be qualified – is test and treat not being implemented in these facilities/in Malawi?

- Line 39-40. This seems to belong to the conclusion.

- The abbreviation MSM, unless used twice or more, may not need to be included - consider leaving only as ‘men having sex with men’.

- Table 1 revision -please consider moving the absolute number (N=133) from the middle column to percentages column; and have % (N=133); this might read better, if acceptable to the authors.

- Table 1: Please also insert a line in the left column, dividing the age and marital status cells

- Line 771: I suggest writing ‘pharmacy’ in small caps

- Table 1: Please align all numbers appropriately – as is done in the rows for education, and uptake of ARVs, for example

- Table 1. “HIV testing’ is written twice

- Discussion – Please review and consolidate any points likely repeated between para 1 and para 2; I see what appears repeated reference to integration. Kindly check again

- The reference to hegemony masculinities - Please confirm that this is the term intended, and that it is how the cited author names it?

- Table 2: Last line, please remove ‘on’ after ‘emphasizes’

- Table 2: Please remove capital letters where these are not needed in the table text

- Please also remove capital letters where they are not needed in headings and sub-titles

Regarding the following query from the authors, please see Academic Editor’s recommendation.

- Reviewer 2: Also, is it relevant to know if the participant is infected with HIV or not? And are they honest about it? I would take that out of the descriptions of the quotes as done in line 380. Response: This kind of identification was brought it after round 1 review. We now need further direction whether this has to be dropped or left as it is.

- The Academic Editor reviewed the authors’ response in the first revision and understood that the authors felt including HIV status provided context to the quotes. It is suggested that the authors may retain HIV status but indicate (maybe in the methods section) how this status was determined and any possible limitations, e.g. whether they are confident the HIV status report is authentic.
---

## [Author Response · Author response to Decision Letter 2]

29 Jan 2021

Dear Editor,

PONE-D-20-08357: “Dispense Antiretrovirals daily!” Restructuring the delivery of HIV services to optimize antiretroviral initiation among men in Malawi.

We are thankful for yet another review over our manuscript which has not only strengthened the message and has highlighted the message we aim to communicate to our readers. 

Please find below the responses to the queries raised:

1. Line 33-35: Please remove capital letters in the phrases ‘in-depth interviews’, ‘focus group discussions’, ‘health care workers.’

Response: This has been corrected. Refer to lines 33-35.

2. Line 36: Please remove ‘were’.

Response: This sentence has been revised and now reads as follows: We digitally recorded all the data, transcribed verbatim, managed using NVivo, and analysed it thematically. This is reflected in lines 35-37

3. 

- Please check all places where ‘Initiation of antiretroviral drugs’ or similar is mentioned. I would presume it is ‘initiation of antiretroviral treatment’ rather than initiation of drugs.

4. Response: Thanks you for this insight, it has been corrected wherever applicable in our manuscript.

5. Please define ‘ART’, ‘ARVs’ at first use, including in abstract

Response: This has been done 

6. Line 39: Please check the sentence in abstract “Restructuring the delivery and conduct of ART clinics”. Do you intend to imply that ART clinics are delivered?

Response: Thanks you for this attention to detail as it has clarified the sentence. It now reads as follows in lines 39-40:

Restructuring the delivery of antiretroviral (ARVs) treatment and conduct of ART clinics is key to optimizing early initiation of treatment among heterosexual men in Blantyre

7. Line 43-44. Consider restructuring/punctuating the sentence to enhance logic and ideas flow. It is not easy to understand specifically how ‘removing of other structural barrier’ links into the whole sentence.

Response: We have revised the presentation and it reads as follows in lines 40-47:

The areas requiring restructuring included: Clinic days by offering ARVs daily; Clinic hours to accommodate schedules of men; Clinic layout and flow that preserves privacy and establishment of male-specific clinics; ARV dispensing procedures where clients receive more pills to last them longer than 3 months. Additionally there is need to improve the packaging of ARVs, invent ARVs with less dosing frequency, and dispense ARVs from the main pharmacy. It was further suggested that the test-and-treat strategy be implemented with fidelity and revising the content in counseling sessions with an emphasis on the benefits of ARVs.

8. Line 45-46: ‘Implementation of test and treat strategy’. I think this phrase could be qualified – is test and treat not being implemented in these facilities/in Malawi?

Response: We have clarified the sentence and it now reads as follows:

It was further suggested that the test-and-treat strategy be implemented with fidelity and revising the content in counseling sessions with an emphasis on the benefits of ARVs. Refer to lines 45-47

9. Line 39-40. This seems to belong to the conclusion.

Response: We are proposing to maintain this sentence as it is more of a topic sentence that sets the platform for presenting the results. I tried to remove it and felt that the results section started without a better introduction.

10. The abbreviation MSM, unless used twice or more, may not need to be included - consider leaving only as ‘men having sex with men’.

Response: This has been noted and has been removed.

11. Table 1 revision -please consider moving the absolute number (N=133) from the middle column to percentages column; and have % (N=133); this might read better, if acceptable to the authors.

Response: This has been done.

12. Table 1: Please also insert a line in the left column, dividing the age and marital status cells

Response: This has been done

13. Line 771: I suggest writing ‘pharmacy’ in small caps

Response: This has been done

14. Table 1: Please align all numbers appropriately – as is done in the rows for education, and uptake of ARVs, for example

15. Response: This has been done

16. Table 1. “HIV testing’ is written twice

Response: The repeated words have been deleted

17. Discussion – Please review and consolidate any points likely repeated between para 1 and para 2; I see what appears repeated reference to integration. Kindly check again

Response: This has been reviewed and it was not easy to note the stated repetition, however, we have clarified in an area where we are extending on integration. The revision is as follows in lines 526-527.

As alluded to earlier, integration of services would be a step towards the implementation of flexible operating rules

18. The reference to hegemony masculinities - Please confirm that this is the term intended, and that it is how the cited author names it?

Response: The statement has been revised to avoid misrepresentation on our part and it reads as follows: These male-friendly spaces will also diffuse the notions of viewing oneself as “the real man” resilient to illnesses which inadvertently impedes ART initiation. Refer to lines 560-561.

19. Table 2: Last line, please remove ‘on’ after ‘emphasizes’

Response: This has been done.

20. Table 2: Please remove capital letters where these are not needed in the table text

- Please also remove capital letters where they are not needed in headings and sub-titles

Response: The manuscript has been revised and all unnecessary capitalisations have been revised.

21. Regarding the following query from the authors, please see Academic Editor’s recommendation.

- Reviewer 2: Also, is it relevant to know if the participant is infected with HIV or not? And are they honest about it? I would take that out of the descriptions of the quotes as done in line 380. Response: This kind of identification was brought it after round 1 review. We now need further direction whether this has to be dropped or left as it is.

- The Academic Editor reviewed the authors’ response in the first revision and understood that the authors felt including HIV status provided context to the quotes. It is suggested that the authors may retain HIV status but indicate (maybe in the methods section) how this status was determined and any possible limitations, e.g. whether they are confident the HIV status report is authentic.

Response: We have noted this advice and kindly note that we provided the means of ascertaining HIV status in the methods section.

Sincerely,

Linda A. Nyondo-Mipando RNM, PhD (Corresponding Author)

---

## [Editor Report · Decision Letter 3]

8 Feb 2021

“Dispense Antiretrovirals daily!” Restructuring the delivery of HIV services to optimize antiretroviral initiation among men in Malawi.

PONE-D-20-08357R3

Dear Dr. Nyondo-Mipando,

We’re pleased to inform you that your manuscript has been judged scientifically suitable for publication and will be formally accepted for publication once it meets all outstanding technical requirements.

Kind regards,

Jeremiah Chikovore

Academic Editor

PLOS ONE
---

## [Editor Report · Acceptance letter]

11 Feb 2021

PONE-D-20-08357R3 

“Dispense Antiretrovirals daily!” Restructuring the delivery of HIV services to optimize antiretroviral initiation among men in Malawi. 

Dear Dr. Nyondo-Mipando:

I'm pleased to inform you that your manuscript has been deemed suitable for publication in PLOS ONE. Congratulations! Your manuscript is now with our production department. 

Kind regards, 

on behalf of

Dr. Jeremiah Chikovore 

Academic Editor

PLOS ONE